# Virtue Ethics and the Ecological Self: From Environmental to Ecological Virtues

**Gérald Hess**

Institute of Geography and Sustainability, University of Lausanne, 1015 Lausanne, Switzerland;
gerald.hess@unil.ch

**Abstract:** This article examines how a non-anthropocentric virtue ethics can truly avoid an anthropocentric bias in the ethical evaluation of a situation where the environment is at stake. It argues that a non-anthropocentric virtue ethics capable of avoiding the pitfall of an anthropocentric bias can only conceive of the ultimate good—from which virtues are defined—in reference to an ecological self. Such a self implies that the natural environment is not simply a condition for human flourishing, or something that complements it by adding the proper good of animals, organisms or ecosystems. Fulfilment is not that of a human self, but that of an ecological self: the natural environment or nature is not an external but an internal good. Therefore, the virtues or character traits that such an ecological self must nurture and develop leads us ultimately to distinguish—without opposing them—three different forms of virtue ethics applied to the environment, depending on whether it is anthropocentric or non-anthropocentric and whether nature is considered extrinsically or intrinsically. Such distinctions are also crucial to determine how we conceive of the political community and the collective goals that virtuous citizens assign to it (for instance, to preserve biodiversity, to tackle climate change, and so on).

**Keywords:** ecological self; human flourishing; ultimate good; ecological virtue; environmental virtue; virtue ethics

## 1. Introduction

Since the 1970s, many proponents of environmental ethics have essentially set for themselves the task of justifying duties towards nature based on the intrinsic value of nature itself [1]. For over 40 years, they have been calling for a moralisation of the human relationship with the environment by means of obligations and prohibitions. The pragmatist approach objected that they were wrong to raise theoretical problems (for example, regarding the intrinsic value of nature). Hence, in so doing, they would distract from the essential task of seeking a consensus on the urgent environmental policies that are the responsibility of governments and prevent an effective response to the ecological crisis. So far, the moralisation has not yet materialised at the political level[1].

Based on a virtue ethics approach, Philip Cafaro [3] notes that environmental ethics has neglected another ethical question, which is just as important as that of human duties and responsibility toward nature: the question of how to achieve the quest for the good that a virtuous person chooses to pursue. For virtue ethics, certain character traits or dispositions of a person (virtues)—to which are added representations and affects—motivate her or him to act in a particular way, as long as these traits are stable, are the result of a deliberate choice and contribute to the person's self-fulfilment [4]. Such an approach focuses traditionally on character traits such as courage, benevolence, temperance, etc., but can also aim at an action insofar as it is evaluated according to good, evil and certain dispositions (or character traits). In so doing, it seems better able to take into consideration the motivation of the agent than deontological or consequentialist forms of ethics which judge the morality of an action according to norms or objective reasonings to be followed. Thus, one may ask:

would a virtue-based approach be more effective than more traditional forms of deontology or consequentialism?

A number of proposals for an environmental virtue ethics have emerged to tackle this issue over the last twenty years [5–8]. But the renewed interest in virtue ethics for environmental issues also raises a dilemma of principles. On the one hand, by seeking to formulate moral principles and duties towards nature, environmental ethics had opened up a way out of the moral anthropocentrism that characterises classical ethics. On the other hand, virtue ethics seemed to head back to a fundamentally anthropocentric ethics since it is focused on human flourishing or happiness (*eudaimonia*) that constitutes the ultimate good—that good which is no longer a means to another end but an end in itself—namely, a good, quality life (*eû zên*) [9] (p. 5 [1095a 17–18]; p. 11 [1097b 1–2]). It follows that a perspective on (non-anthropocentric) environmental ethics based on (anthropocentric) virtue ethics seems hardly compatible with its initial intention [10]. In this article, I will look at several ways of dealing with this dilemma.

To this end, I will distinguish two ways of looking at the relationship between virtue ethics and the natural environment.

- Extrinsically. The environment is a condition either for the exercise of virtues or for human flourishing; it is not part of the human flourishing itself. Moreover, one could possibly extend virtue ethics to other goods that are not strictly related to human happiness. In this case, one could speak of environmental virtues that contribute to the "good life" of an individual, whether human or non-human, such as temperance, for instance.
- Intrinsically. The environment is part of the human good and not just a condition of that good. This means that humans are not external to nature; they are part of it in such a way that, by virtue of an intrinsic relationship between humans and the environment proper to an ecological self, the flourishing of humans is neither separate from nor independent of the flourishing of nature. In other words: nature is not external to our existence; it constitutes it. Therefore, in this case the ultimate good of a virtue ethics is not the flourishing of human self but that of an ecological self.

I will develop this distinction in the next three sections. In Section 2, I will first clarify the characteristics of virtues ethics. In Section 3, I will examine how an extrinsic relationship translates into environmental virtues and show the limits of this conception. In Section 4, I will explore how an intrinsic relationship allows us to reach beyond these limits based on the notion of the "ecological self", a notion already defended by the deep ecology of Arne Næss and by the ecofeminism of Val Plumwood or Freya Mathews. I will reckon that certain virtues are fundamentally ecological and not simply environmental, because they are ascribed, strictly speaking, to an ecological identity. That is, when the relationship with the environment remains extrinsic, the virtues developed are at best environmental virtues; when the relationship with the environment becomes intrinsic, by returning to the lived experience of the world, the virtues are truly ecological. Finally, I will conclude that this proposal can overcome the anthropocentric bias underlying environmental virtue ethics. What is at stake in identifying this anthropocentric bias is not simply to emancipate ourselves from the *centred* character of moral anthropocentrism, which limits itself to taking account of humans and humans alone. It is a question of overcoming the *centric* character of moral anthropocentrism, which reduces the world to the human perspective alone.

## 2. The Characteristics of a Virtue Ethics

Generally speaking, a virtue ethics does not seek to answer the question, "What should I do?" but rather the question, "What kind of human being do I want to become?" The virtuous person strives towards a life that she or he considers good, a life of quality (*eû zên*) or of flourishing (*eudaimonia*). This requires the exercise of certain dispositions in the pursuit of excellence, namely virtues. These practical dispositions are called virtues when they are sufficiently stabilised to be perceived by an agent as motivating his action. In this respect, virtues define the character traits of an individual: goodness, generosity, humility,

etc. But not every character trait is a virtue. For it to be a virtue, it must be related to the ultimate good and aimed at for its own sake, i.e., in the aim of what the agent considers to be a flourishing life. Virtues are therefore character traits in the pursuit of excellence, in the sense that they contribute to and are even a part of happiness and a fulfilled human life. In virtues, beliefs, affective orientations, perceptual dispositions and behavioural tendencies are inextricably linked [11] (p. 14–15). These general considerations require two clarifications. The first concerns the relationship between virtues and duty. In truth, an ethics of virtues is not opposed to an ethics of duty, in the sense that in its own way it also aims to act, but with reference to virtues and not according to an obligation or moral principle. The second point concerns the ultimate good, i.e., human flourishing, a fulfilled human life. While it is true that a character trait is a moral virtue because of its contribution to human flourishing, virtue does not relate strictly to the virtuous agent's own flourishing; it also concerns other humans and, possibly, other non-humans.

For Aristotle, to whom we owe a systematic study of the ethics of virtues, a good or fulfilled life depends on the effective performance of what is inherently human. Just as a "good" eye depends on the eye's ability to see without distortion, the "good" human being depends on the human's ability to act virtuously. For Aristotle, what defines a well-functioning human being is the exercise of reason [9] (p. 103–107 [1139a and b—1140b 1–10). Indeed, the human good necessarily has to do with what makes us human, and what distinguishes us as humans from other species is the use of reason. Thus, it would follow that it is reason that enables us to live a good human life [12] (p. 6). To be virtuous, and therefore to live a flourishing life, as Philippa Foot puts it, rational will must be translated into purposeful, voluntary actions [13] (p. 66–67, 69–70).

Today, however, it seems increasingly difficult to define the highest good in reference to a human nature. Irene McMullin distinguishes two ways of conceiving of the ultimate good [11] (p. 24). The first way is from a subjective perspective in terms of the agent himself: a fulfilled human life is what each person decides it to be; fulfilment then comes down to the agent's evaluation of her or his experience, independently of its content, which is no longer relevant in terms of defining that good. This first perspective responds to a modern vision of a plurality of conceptions of what is good but runs counter to the widespread—and indeed self-evident—idea that there are basic goods on which everyone can agree, such as healthy food, a good education, a social life, and so on.

The second perspective sees the human good not from the subjective perspective of the agent, but from an objective point of view. In this case, the good is considered in relation to biological functions such as survival, the continuity of the species, a pain-free state and the successful functioning of the social group [14] (p. 153–154). The definition of what is good must then be based on physiology, biology and ethology; this is, strictly speaking, the naturalistic point of view of science. But to leave it at this point would be to reduce human flourishing to externally observable traits of human reality, while overlooking the fact that these traits are—at least in part—experienced by human beings, i.e., that each human being appropriates them and lives them from her or his own perspective.

This is why, rather than opposing the two conceptions—subjective and objective—of a good or fulfilled life, McMullin suggests "reconceptualizing the self" [11] (p. 30). The human self whose fulfilment we seek is not simply myself; it includes the self of other humans and the self of the human community to which I belong, i.e., its values and ends. A person's character cannot be assessed simply by observing him or her from the outside as an organism with objectively defined ends (survival, continuity of the species, etc.). But the subjective perspective seems just as flawed if the agent is simply judged from the angle of his subjective experiences (positive or negative), with himself and the others, independently of what constitutes his biological basis, of scientific knowledge in general and of the community to which she or he belongs [11] (p. 31). Thus, the self is a reality that must be approached in its unity, both as a subjective, lived experience and, so to speak, as an "object".

Ultimately, self-realisation in the world, from the point of view of happiness or a fulfilled life, must be approached, according to McMullin, from three complementary perspectives. From a first-person perspective, it operates on the basis of what is specific to me (my identity, what defines me, my unique perspective). From a second-person perspective, it includes everything that stems from my relationship with others (their needs, vulnerability, interests, etc.). From a third-person perspective, self-fulfilment in the world is motivated by scientific knowledge and by certain values shared by the community to which I belong (freedom, equality, friendship, and so on) [11] (p. 37; 40–64).

This contemporary redefinition of the good or flourishing human life both renews and extends the classical—Aristotelian—tradition of virtue ethics. Yet it is striking to note that this redefinition in no way refers to the natural environment as an aspect of self-realisation in the world. The natural environment as moral patient is absent from McMullin's discussion. In this respect, the concepts that define virtue ethics seem to lend support only to a moral anthropocentrism[2].

### 3. Environmental Virtues: The Extrinsic Relationship between Virtue Ethics and the Environment

Ronald Sandler's thinking is one of the most accomplished contributions to date that addresses this blind spot and endeavours to take the natural environment into account in virtue ethics [7]. In his approach, he examines the different ways by which the environment can contribute to a virtuous life. Firstly, the environment can be seen as a condition for human flourishing (moral anthropocentrism). Secondly, it can be considered for its own good (moral non-anthropocentrism).

### 3.1. Moral Anthropocentrism

According to Sandler, the anthropocentric perspective can make sense of the environment in virtue ethics in two ways: as a condition for a person to be virtuous, or as a condition for human happiness[3].

In the first case, a healthy environment would be necessary for the exercise of virtues. This would mean, for example, that a degraded environment would prevent us from being virtuous. But if this degradation were compensated for by technological innovation or other artificial goods, it would no longer be a problem. On closer examination, this argument is not convincing, because it is difficult to objectively assess the threshold at which the ecological conditions would cease to enable virtue, if such a threshold even exists. Does a severely degraded environment really prevent gratitude, love or frugality, for example? Experience seems to show the opposite: it is often in difficult times, such as war, that certain virtues—frugality or solidarity, for example—motivate people's behaviour. Furthermore, the latter part of this argument implies that technology could, if available, easily compensate for degraded environmental conditions. This presupposes the utilitarian meaning attributed to the environment, but ignores all the very real values—aesthetic, cultural, patrimonial, and so on—that individuals or communities ascribe to their environment, and that commit those people to protecting it, regardless of its utility.

But—and this is the second case—it is easy to see the environment as a condition for human flourishing. There is no doubt that in many situations the quality of the natural environment contributes to a good human life. A healthy soil means that we can produce good quality food in sufficient quantity; unpolluted air means that we can maintain good health, and so on. A healthy environment also encourages social cohesion and cooperation between communities. The inverse is also true: the virtues of justice and concern for the people in our community and between communities, for example, help to promote good environmental conditions.

The two ways by which virtue ethics can make sense of the environment with an anthropocentric perspective assume that there is an extrinsic relationship to be found between the environment and humans.

When nature is seen as a simple condition for a good human life, such a vision remains dependent at worst on an ontological dualism between man and his environment, and at best on an epistemological dualism between the knowing subject and the object to be known. Let us begin with the objection posed by ontological dualism. Ontological dualism implies that the human being stands out from the environment, a classic opposition established by modernity between a material reality (nature) and a spiritual reality (culture), harshly criticized today by anthropologists[4]. This ontological dualism provides a justification for a moral dualism. If the environment is no more than a means to the end of achieving human good, it has only an instrumental value in the service of human flourishing, whereas the latter is intended for its own sake, and therefore has its own intrinsic value. The difference between the instrumental value of nature and the intrinsic value of human life duplicates an ontological difference between an environment that is fundamentally inert and devoid of value in itself and human beings who are both the source and the repository of value.

Admittedly, the conception of the human being standing out from nature has been undermined by the advent of scientific ecology and the theory of evolution, among others, which state that human beings actually belong to nature and have evolved from it. However, epistemological dualism raises a second objection. From the point of view of scientific knowledge, while humans are ontologically reintegrated into nature, they nonetheless remain outside it as knowing subjects. The epistemological model of a disembodied knowing subject, external to the object it seeks to understand, guarantees objectivity in science. It provides the basis for a third-person approach to the world and has structured the development of scientific knowledge until now. This second objection is more difficult to unpick in the context of virtue ethics and is usually not given sufficient attention. This can be further clarified by considering the way Sandler tries to escape the anthropocentric bias in virtue ethics.

### 3.2. Moral Non-Anthropocentrism

Without departing from scientific naturalism, Sandler believes that a virtuous life is not limited to the pursuit of human flourishing in the sense developed above. Virtue ethics, he says, is also able to take into account the natural environment, not only as a condition for human flourishing but as an end in itself. He explains that what makes certain character traits virtuous need not necessarily be a function of human happiness alone, for what makes a character trait a virtue has to do also with ends other than human happiness, insofar as these virtues too are considered ends in themselves. Thus, if it is acknowledged that animals and organisms have their own immanent good independent of that of humans (the satisfaction of an animal's own desires or needs, for example), certain character traits can be seen as virtues in relation not only to human ends (happiness), but also to desires or needs of animals, to ecosystemic health and even to ecological integrity [7] (chap. 3).

Sandler's proposal therefore seeks to extend virtue ethics to considerations that are no longer solely concerned with the human good, but with a good that is specific to the natural environment or to its elements. In this respect, an ethics of virtues would become compatible with the attempts of environmental ethics to overcome the moral anthropocentrism underlying classical ethics. Some aspects of the environment are then no longer simply a condition for human flourishing; they take on a properly moral value, based on their own good. Temperance, for example, which traditionally aims to control our desire for pleasure in areas such as food or sexuality, would become an appropriate character trait for moderating the consumption of natural resources, where this consumption is detrimental to animal welfare, the well-being of certain organisms or the ecological integrity or health of an ecosystem. Hence the virtue of temperance, initially conceived in relation to human happiness and flourishing, now has a new goal: the promotion of the good of certain aspects of the environment. Thus, here we see a traditional virtue which, when applied to environmental issues, becomes an environmental virtue.

Applying virtue ethics to the environment is undoubtedly useful in everyday practice. However, as long as the emphasis is on the character of a person in relation to human

happiness and the good of the environment, this confrontation will meet numerous conflicts of interest in which human happiness is very likely, depending on the circumstances, to relegate what is good for the environment to second place. For example, the benevolence that encourages me to visit my sick grandmother, who lives thousands of miles away, contrasts with the temperance that encourages me to avoid long plane journeys. Or, within a virtue such as love, for example, love for my sick grandmother is opposed to my love for nature, a love that strives to avoid an act that is harmful to the environment (air travel). No doubt benevolence will win out here over temperance, or love for my grandmother over love for nature. As long as the moral agent is considered to be a human subject (i.e., a moral subject), according to the logic of virtue ethics, the good of non-human patients will often be evaluated in favour of the human patients who are human subjects too and therefore moral subjects. This is because an environmental virtue ethics remains subject to an epistemological subject–object model. Such a model underlies the structure of the ethical relationship between moral subject and moral patient. What makes a human subject a moral subject is determined, as Philippa Foot shows, by the fact that the human subject possesses a rational will. Naturalism may strive to define objectively, from the outside, the properties of the object likely to make it a moral patient (for example, the happiness of other human beings, the well-being of animals, the good of organisms, etc.) and the properties of the subject likely to make it a moral agent (for example, the rational will of the human). However, decision in the sphere of moral assessment and behaviour will ultimately be determined by the human will: the rational will always choose itself over that which differs from it. This preference ultimately results from the subject–object structure underlying naturalism, which defines what is a human subject (a moral subject) as opposed to what is merely an object—or, at most, a moral patient.

To address this anthropocentric bias, we need to turn to a level that escapes the human subject/object or moral subject/moral patient structure. This level is that of a lived experience which, as we shall see, has not yet fixed on an intentional relationship to an object or to a moral patient.

### 3.3. The Remaining Anthropocentric Bias

In order to fully grasp the sense of this objection, we need to clarify what anthropocentrism means. An initial distinction can be made between epistemic anthropocentrism and moral anthropocentrism. The anthropocentric bias referred to above does not fall under moral anthropocentrism, since, as we have seen, environmental virtue ethics is apt to considering non-human moral patients. The objection therefore concerns epistemic anthropocentrism (see Figure 1, below). But this expression can remain a relatively vague concept and needs to be clarified.

| 1<br>Kind of Anthropocentrism | Epistemic anthropocentrism | | | |
|---|---|---|---|---|
| 2<br>Human Mind | Mental dimension<br>Psycho-physiological functionnality | | | Physical dimension<br>Physical body, brain |
| 3<br>Human Subjectivity | Ontological Perspective<br>(in first person) | | Epistemic Perspective<br>(in third person) | |
| | Non-objectified | Objectified | | |
| | | Subject,<br>self-consciousness | Reason, affectivity, belief, interest | |
| | Lived experience | Personal identity | | |

**Figure 1.** Epistemic anthropocentrism.

In a first approximation, epistemic anthropocentrism means that human consciousness must be presupposed for knowledge and evaluation of nature. Without human conscious-

ness, there could possibly be a reality, but it would be perceived and thought of differently than it is by the human species. The idea underlying epistemic anthropocentrism has two dimensions, physical and mental. The physical aspect consists of the physical body by which reality is apprehended, in particular the brain. The mental aspect, on the other hand, comprises the psycho-physiological functions that the body and brain enable (cognition, affectivity, conation, consciousness). These two aspects together form what we might call the mind[5]. On a third level of epistemic anthropocentrism, the mental aspect of the mind can be approached from a strictly epistemic angle, i.e., from an external, third-person point of view, as reason, affects, interests, beliefs, desires, and so on. However, it can also be approached from an ontological angle, from an interior, first-person point of view. It is then a conscious subject, aware of himself, who thinks and acts rationally, exercises his will, has interests, desires, etc., in relation to his social and natural environment. Viewed from both inside and outside, the mind comes to constitute, so to speak, the personal identity of a human being or, in other words, the subject (knowing subject, moral subject, etc.).

Having said that, can we reduce the mind to a personal identity? In fact, this reduction can only result from an objectification of the subject, whether this objectification is done by himself or by other subjects. For apart from a constituted self—that is, insofar as we strive not to objectify the subject—lived experience seems, on the contrary, to attest that it is much more fluid and diffuse, without precise limits. It suggests relationships rather than objects or subjects. Co-extensive with experience, the mind is then no longer limited to the human subject and his body; it is still neither truly subject nor truly object, but traverses what surrounds our lived bodies in a multitude of ways through the beings we encounter (humans, animals, plants, ecosystems, etc.)[6]. The anthropocentric bias then refers precisely to the ignorance or exclusion of this underlying—non-objectified—dimension of mind that precedes any representation of humans and non-humans as subjects or objects, of moral (human) agents and of moral (human and non-human) patients.

As Figure 1 shows, anthropocentrism is hardly questionable at the first and second levels [epistemic (1), human mind (2)]. On the other hand, in order to avoid falling into an anthropocentric bias, we must—at the third level, i.e., human subjectivity (3)—not reduce our understanding of the concept to that objectified part of the subjectivity that lets its non-objectified part slip away. For it is within this non-objectified part that reside the meaningful relationships between humans and non-humans that are likely to motivate action, before they crystallise and become fixed in a personal identity through character traits.

For the moment, we can therefore draw the following conclusion. In order to avoid an anthropocentric bias in the ethical evaluation of a situation based on virtues (and vices), it is necessary to reconsider the subject–object model and also to relativise the scope of naturalism in the definition of the ultimate good. For while naturalism ensures objectivity in the definition of the human and non-human good, it is also naturalism that justifies the human subject in his exclusive role as a moral agent in relation to moral patients. Thus, before any objectification by science (subject/object) or morality (agent/patient) in a third-person perspective, we must first remember that we are dealing with lived relationships between selves or agents (human and non-human) in a first-person perspective. Such a first-person perspective avoids the pitfall of the epistemological dualism underlying naturalism; it allows access to the *relational* dimension of lived experience which is an experience *with* nature and no longer an experience *of* nature.

In other words, self-realisation in the world can no longer be seen simply as that of a human subject or agent in relation to objects or patients (human and non-human). Before any recourse to naturalism, before any objective consideration, it must be approached, first and foremost, from a subjective perspective that provides access to the natural environment from the inside, without remaining outside it. The promotion of a moral non-anthropocentrism in virtue ethics, such as Sandler's, does not escape an anthropocentric bias. The latter is ultimately based on an epistemological dualism in which the relationship between humans and their environment remains extrinsic.

This leads to shift the emphasis from environmental virtues to ecological virtues. As we shall see, in an ethics of ecological virtues, the good life, or the good to which these virtues are attached, needs to be considered as that of an ecological self, i.e., the self that must be considered from a first-person perspective.

### 4. Ecological Virtues: The Intrinsic Relationship between Virtue Ethics and the Environment

*4.1. A Relational Ontology, beyond the Epistemological Subject–Object Model*

Returning to a subjective, first-person perspective involves an awareness that the object of experience does not exist independently of the intentional relationship of the subject established through that subject's experience. For example, when I see a tree through the window, I may become aware that the tree does not exist in itself, but that it appears to me in the perception I have of it, i.e., in my perceptual relationship with the tree. However, if the subject is in an intentional relationship with the environment, the latter can be seen in two different ways. First, it can be seen as a spatially constituted object (empirical or conceptual). For example, a tree can be seen as a cherry tree whose tasty fruit I eat in summer. We are dealing with a relation of meaning, but of a meaning that has already been formed. Here, the subject–object model continues to permeate the relationship with nature.

The second way aims to get rid of the subject–object structure. The subject's intentional relationship with the environment is then no longer with an already constituted object, but with an object as it appears in the present experience. It is that of an *appearance of the object*, of its genesis in experience, *before* the division into subject and object, i.e., below the objectifying and spatialising subject–object structure [18] (p. 103–104). In this case, the environment does not pre-exist the *event* through which I encounter it In the life-world. Indeed, through my body, I belong to the life-world. I constitute this world as much as it constitutes me. Now, this co-constitution is event-based and precedes both the "subject" and the "object" [19] (p. 726–727); it relates to the temporal structure of consciousness. It follows that, in this respect, the value of the environment belongs neither to the realm of physical reality (object) nor to the realm of mental reality (subject); it is neither subjective nor objective. It comes from a reality—the life-world—that is fundamentally relational: a field of consciousness that pre-exists a valued object and the subject who values it. To take the previous example, the tree in my garden is experienced in the event of the shimmering branches of the cherry blossom that I see waving in the spring breeze, in the event of the deep red colour of the cherries when they are picked up, or in the event of their tender, sweet flesh when I taste them.

In the latter case, the "de-objectification" is radical. This de-objectification is radical in the sense that it frees us from the epistemological subject–object model and definitively turns our backs on an ontology that conceives of the entities of the world as separate from one another. This other, relational ontology is neither an ontology of the object (materialism) nor of the subject (idealism), but of a relational field in which the value of nature is above all an event. Both the evaluator and what is being evaluated depend on this event, which is the origin of practical dispositions—character traits or virtues—that motivate us to act.

Consequently, the natural environment is not simply a condition of the human good but is itself an aspect of the good. In other words, it is a constituent of flourishing, of a realisation that is not exclusively that of the human self but encompasses everything with which human beings are in relationship: other humans, but also nature (animals, living things, soil, air, natural things like rivers or mountains, landscapes, and so on). The supreme good that an ecological virtue ethics must pursue is thus not that of a human self, but of a self that defines itself in relation to its environment. As we have seen, the character traits that make up the virtues, according to Sandler, are not only a function of human good (happiness), but also of ends immanent to non-human natural entities (their own good). However, if the self is not what we think it is, then the good, flourishing life is not what we think it is either. A self extended to the environment is an ecological self, not a human self. It is not defined as an entity separate from its environment, but as a

node of relationships within a relational network. The realisation of such a self means the crystallisation of a knot of relationships constituting the environment, in which true happiness is achieved. It follows that a virtuous human being must be judged according to a renewed understanding of fulfilment: that of an ecological self, not a human self.

Before taking a closer look at what the flourishing of an ecological self means, let us briefly note some advantages of an ecological ethics of virtue, in other words, a non-anthropocentric virtue ethics, which the above development makes it possible to highlight. Compared with an environmental virtue ethics, such as Sandler's, which seeks to extend the scope of certain classical virtues to the good of the environment, an ecological virtue ethics defends the idea of properly ecological virtues. These virtues presuppose an acute awareness of the anthropocentric bias that constantly threatens our day-to-day practice and that Sandler's approach cannot avoid. In an ecological virtue ethics, however, awareness of an anthropocentric bias invites us to return again and again to the events of our lived experience of nature, within which our multiple relationships to the environment are born and renewed.

Moreover, ecological virtues are not limited to natural entities that have a proper good, a good in and for itself, but can allow us to take into account all the constituent elements of the natural environment, without having to worry about the (theoretical) question of whether such and such an entity actually has its own good or ecological integrity. An ecological virtue ethics is able to do this because it is based on a relational ontology—where the entities of nature (including ourselves) exist through the relationships they enter into with each other—and not on a naturalistic ontology in which the entities exist separately from and are independent of each other.

An ecological ethics of virtue finally makes explicit a deeper living dimension of human experience than that of perception, which motivates and guides our actions and, more generally, our behaviour. In the environmental virtue ethic model, on the other hand, virtues are applied to objectified entities which, unlike the relationships we have as humans with other humans, leave little room for the affectivity inherent in the experience that permeates our motivations. By escaping epistemological dualism, the ecological self can also hope to escape the anthropocentric bias.

### 4.2. The Good Life, or the Ultimate Good of an Ecological Self

The ecological self is not a new idea. It was proposed in the 1980s by the philosopher Arne Næss [20] and [21] (p. 171–183), developed later theoretically by Freya Mathews [22] and then taken up again, with certain nuances, in the ecofeminism of Val Plumwood, among others [23] (ch. 6). In the philosophical tradition of phenomenology [24], let us say that the ecological self refers, in a first approximation, to a relational self whose realisation includes, among its own primary ends and through different modes of participation, the fulfilment of other beings on earth. In this context, I examine how to make sense of the three-dimensional model that McMullin uses to define human flourishing (see above).

Adapted to the concept of an ecological self, the first-, second- and third-person perspectives can be replaced, respectively, by those of interiority (a relationship to oneself), exteriority (a relationship with nature or the environment in general) and objectivity (scientific concepts validated intersubjectively).

- First-person perspective. Considered primarily from the point of view of interiority, the realisation of the ecological self—its flourishing or fulfilment—no longer operates on the basis of a personal and social identity of its own, but on the basis of a decentring of oneself, or, so to speak, a "dying" to oneself. This implies a return to the impersonal layer of my existence, to that of the lived body, the body that feels and lives in the space-time and temporality of consciousness. It is through such a body that we can grasp the otherness of natural beings and act virtuously towards them at the very heart of this relationship with the other. The self refers not to the *ego* but to a dimension of being deeper than that of the beliefs, desires, interests and feelings with which I usually identify. The decentring of oneself therefore concerns not the self but what

defines the self socially and personally as an *ego* who does not reflect on and transform that self.

- Second-person perspective. From the point of view of exteriority, the realisation of the ecological self includes everything that can be accessed through the various ways of participating in nature. This participation can take various forms. For example, through empathic participation with an animal, I am able to respond to its sensitive, affective and even, in some cases, conative life. Meanwhile, in a form of participation that I describe as 'enactive' (because of the finality immanent in every organism that produces its environment), I am able to respond to the vital activity of an organism, as it manifests itself in metabolism (feeding, breathing) and movement (desiring). And in a 'trajective' participation (in which I project my body into the environment), I can grasp, as part of a human community, the characteristics of my environment and of the particular space-time within which my existence takes place: the climate, the aridity or fertility of the soil, the presence or absence of an animal or plant population, the singular beauty of a landscape, and so on. And, in so doing, I can also respond to the particularities of the natural environments in which other human communities live.
- Third-person perspective. Finally, from the perspective of objectivity, the flourishing of the ecological self is also motivated by the knowledge developed over time through scientific research, which, in the context of ecological awareness, aims to gain a better understanding of the non-human animal, the living and the plant world in particular, so as to have a more accurate view of ecosystems and, more fundamentally, a more complete understanding of reality. As we can see, the third-person perspective of scientific naturalism has not been abandoned. But it is relativised in relation to a first- and second-person perspective.

An ecological virtue ethics can only be developed on the basis of an ecological self. Nevertheless, in the context of this article, it is not possible to identify and describe ecological virtues any further. In the final part of this section, I shall confine myself to illustrating what an ecological virtue could be on the basis of a well-known figure in environmental thought. Drawing on her personal experience, the philosopher and ecofeminist Val Plumwood insists on the development of a relational ethics, based on our lived participation in the environment.

### 4.3. An Ecological Virtue: Vigilance or Attentiveness

In February 1985, Val Plumwood set off on an excursion to Arnehm Land, east of Kakadu National Park in Australia's Northern Territory. While crossing a river in a canoe in the monsoon rain, she was surprised by a saltwater crocodile. Suddenly, the crocodile pounced on the boat, biting her leg and trying to drown her, but miraculously, albeit wounded, Plumwood managed to escape the predator and flee. With the benefit of hindsight, this traumatic episode became a crucial event in Plumwood's life. It enabled her, she says, to become truly aware, in her own flesh, of the very relative place of the human species in nature, when confronted with a predator. But that is not the whole story.

Afterwards, she asked herself: "Why did I do such dangerous things and not perceive my danger? Why did I not see myself as subject to these kinds of dangers in this place?" [25] (p. 14). When she set out to cross a river infested with sea crocodiles, she thought she was safe. She thought this was her world, a world she knew and was familiar with.

> Yet, as I looked into the eye of the crocodile, I realised that my planning for this journey upriver had given insufficient attention to this important aspect of human life, to my own vulnerability as an edible, animal being. [25] (p. 10)

> One way of answering these questions lies in my background in a certain kind of culture, my background relationship to the land I was visiting and the land of home. My relationship, in other words, to place. I was in a place that was not my own and which was very different from my own place. An important part of place is one's sense of the large predators for placing us. [25] (p. 20)

Plumwood did not realise the risk she was taking by paddling a canoe on a river she knew to be inhabited by crocodiles. Basically, she failed to be vigilant. By failing to pay attention to a habitat that was not her own, she failed to realise that she was sharing it with other animals, not least her own predators. The importance of this sense of "the large predators", or vigilance, is also what Plumwood has in mind when she comes to the end of her story, recalling the forest fires in the south-east of Australia where she lives:

> The southerly change really is Cool. [...] I dig out a sweater; lyrebirds are singing again; grasses greening. [...] The dripping forest feels good now, but I know it's not over yet until we get a lot more rain. [...] You must be able to look at the bush you love and also imagine it as a smoking, blackened ruin, and somehow come to terms with that vision. I am trying to make my house fire-ready, but in the cool moist airstream of the moment I am finding it hard to sustain the sense of urgency and inevitability [...]. But I know I will have to meet the fire monster face-to-face one day. [25] (p. 21)

Vigilance is a disposition that translates into attention to the world around us, which comes to us through our experience of it: the beauty of the nature around us, but also the anticipation of the fire that might destroy it.

The virtue of vigilance, as it emerges from this story, is a truly ecological virtue. For it emerges from an identity that does not see itself as separate from the environment in which it evolves. This self is an ecological self, in the sense that this identity is lived and experienced in relation to the place it inhabits, its climate and the other animal and plant species that live there with it: the wildfires, the humid air of the ocean, the lyre birds and the trees of the forest. It is precisely this intimacy with the natural environment that leads it to develop a vigilant attitude to the dangers of fire.

I think that such a virtue of vigilance, understood from the point of view of an intrinsic relationship with the environment, differs from the meaning it would take on in an environmental virtue ethic. Viewed as part of the development of an ecological self, vigilance here reflects an amplified experience of nature in which the threat of an attack by a predator or a forest fire is placed in the context of a lived proximity to the environment—for example, an acute attention to those who share our habitat or to the singular climate of a region. This presupposes the ability to imagine a reality different from the one we are currently experiencing and to anticipate the behaviour it implies. In an extrinsic relationship with the environment, however, it is precisely this amplification of lived experience closely associated with imagination and anticipation that is lacking. In this case, the anthropocentric bias and objectification of the lived experience encourages us to focus our attention on the predator (which we are going to eliminate) or the fire (which we are going to fight) without seeing them as events inherent in the environment we inhabit and with which we must learn to live.

Through the narrative of Plumwood, the virtue of vigilance illustrates its meaning as an ecological virtue. It differentiates itself from the sense it would take as an environmental virtue within an environmental virtue ethics where the relationship to nature is extrinsic and not intrinsic.

The three forms of virtue ethics applied to the environment can now be distinguished in the figure below (Figure 2) according to moral anthropocentrism or moral non-anthropocentrism in virtue ethics and according to the extrinsic or intrinsic relationship with nature.

Figure 2 shows the differences between the three forms of virtue ethics, depending on whether they are founded in moral anthropocentrism or not and whether they are intrinsic or extrinsic to the environment. An environmental virtue ethics is close to an ecological virtue ethics in terms of their common, non-anthropocentric perspective. But it is close to a classical virtue ethics in terms of the extrinsic nature of the relationship to the environment. Only an ecological virtue ethics is both non-anthropocentric and intrinsically committed to nature. In so doing, it avoids the pitfall of anthropocentric bias that constantly threatens an environmental virtue ethics.

| Moral Anthropocentrism | | Moral Non-Anthropocentrism | |
| --- | --- | --- | --- |
| Classical virtue ethic | Environmental virtue ethic | | Ecological virtue ethic |
| Extrinsic relationship to nature | | | Intrinsic relationship to nature |
| Human flourishing | Human flourishing + the « good » of the environment | | Flourishing of an ecological self |
| Classical virtues (benevolence, love, etc.) | Environmental virtues (temperance, etc.) | | Ecological virtues (vigilance, etc.) |

**Figure 2.** Three forms of virtue ethics applied to environment.

## 5. Conclusions

The aim of this article was to sketch out a response to the dilemma inherent in a virtue ethics that endeavours to take account of the natural environment. I have tried to propose a non-anthropocentric version, an ecological ethics of virtue, although not an environmental virtue ethics. I have tried to show that the individual who operates on the basis of environmental virtues has an extrinsic relationship with his environment in which the good of the environment is added, so to speak, to human happiness from the outside. This individual remains subject to a dualist subject–object epistemological model that is unable to account for the relationships it has with its environment. In this approach, the environment is at worst a mere object, useful or useless in satisfying human needs and desires; at best, it is a moral patient whose value must be taken into account by a human moral agent who himself has a value and who is responsible for the moral assessment of a situation. This is the inevitably anthropocentric bias of this model applied to environmental ethics.

Unlike environmental virtues, ecological virtues are those of a genuine ecological self whose relationship with its natural environment is intrinsic, not extrinsic. In this case, the relationship we have with nature prevents it from being assimilated to a mere object or to something radically different from ourselves, just as much as it avoids reducing it to ourselves [26]. The relationship with nature strives—through participative relationships—to preserve nature in what constitutes it in its own right: its own experience (animals), its life (organisms) and the environments in which human and non-human communities live, which are always unique (ecosystems).

It will be objected that this conceptual clarification may seem futile in practical terms. After all, in everyday situations where the environment is at stake, it is highly likely that a virtuous person animated by environmental virtues will often act in a similar way to one animated by ecological virtues. In fact, it is not a question of pitting the three versions of virtue ethics that I have proposed in this article against each other. Each has legitimacy at some level of our relationship with the environment. From an anthropocentric perspective, classical virtue ethics can be favourable to the environment. Think of benevolence towards non-human animals or, more generally, love of nature. But here the relationship with the environment remains fairly superficial. From a non-anthropocentric perspective, an environmental virtue ethics makes it possible to consider nature or certain aspects of it morally by redefining certain virtues specifically in relation to the environment, such as temperance. Ultimately, an ethics of ecological virtues is one that probes deeply into our relationship with the environment by aiming at the stratum of lived relationships with nature, before any implementation of the subject–object epistemological model. Ecological virtues are also redefined in relation to the environment, but they are redefined on the basis of our lived experience of it, and not on the basis of the properties of objectified natural entities, identified as belonging to their good.

As Plumwood's example of vigilance illustrates, the character traits of an ecological self presuppose a different experience of how humans belong to their natural environment and a different way of perceiving and behaving in it. An amplified experience of nature, such as Plumwood's, takes account of human and non-human interests alike, aiming for the fulfilment of a whole to which a person is aware of belonging. Furthermore, those

differences inevitably have an effect on the way we conceive of the political community and the collective goals we assign to it. This will inevitably determine the contents, the demands and even the radical nature of environmental policies on, for example, the preservation of biodiversity, climate change or energy transition that are put in place democratically and ultimately accepted by virtuous citizens. I think, for example, that political management of a predator like the wolf, based on ecological virtues, is significantly different from management based on environmental virtues. There is always a risk that the anthropocentric bias, coupled with an immoderate objectivation of the environment, will favour a policy that gives more weight to human fulfilment (in this case, that of livestock farmers and mountain farmers) than to that of an ecological self. The latter will consider a priori the legitimacy of wolves living in a territory they share with farmers. This means considering farmers and wolves together as part of the same ecological reality, while taking account of the interests of both sides—thanks above all to the various forms of participation in nature and that ecological knowledge can complement. Wolves are not just livestock predators (objectivation) that need to be eliminated (anthropocentric bias), and farmers are not just livestock owners (objectivation) that need to be protected (anthropocentric bias). In this way, the solution cannot be subsumed to a norm (apart from that of being virtuous) or be the result of objective reasoning; it is adapted to situations that are always singular. And this is something that only ecological virtues or character traits can address.

The task now ahead is to identify which character traits constitute authentic ecological virtues. This can be achieved through a more precise description of the ways in which humans participate in nature[7].

**Funding:** This research received no external funding.

**Institutional Review Board Statement:** Not applicable.

**Informed Consent Statement:** Not applicable.

**Conflicts of Interest:** The author declares no conflict of interest.

## Notes

1    See the arguments in Andrew Light and Eric Katz [2].

2    Aristotelian ethics and the tradition associated with it—which for the purposes of this debate I will refer to as "classical virtue ethics"—do not envisage the human being separated from his environment; however, the external nature within which the human being flourishes is nonetheless a condition for his flourishing, and no moral value is attributed to it. It is an ethics of and for human beings: it concerns them and is addressed to them. We could say that such a moral anthropocentrism is an anthropo*centred* ethic, but not necessarily an anthropo*centric* ethic. Moral anthropocentrism will become anthropo*centric* in modernity with the ontological and epistemological dualism (see Section 3 below).

3    For a discussion of these two arguments, see Ronald Sandler [7] (p. 43–55).

4    See, for instance, Philippe Descola [15].

5    See the very insightful paper of Max Velmans [16].

6    In the field of environmental ethics, see Gérald Hess [17], especially p. 77–83.

7    I would like to thank Marine Bedon for drawing my attention to her useful formulation of the distinction used in Note 2, and Sylvie Pouteau for her comments and her suggestions to the clear structuring of this article.

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
