# Peer review of "Virtue Ethics and the Ecological Self: From Environmental to Ecological Virtues"

_philosophies, doi:10.3390/philosophies9010023_

Round 1

Reviewer 1 Report (Previous Reviewer 2)

Comments and Suggestions for Authors

Thanks for your careful consideration of my previous suggestions

Author Response

Dear reviewer

Thank you once again for taking the time to read my article critically. 
I'm delighted that the changes I've made in response to your comments have met with your approval.
Yours sincerely

Gérald Hess

Reviewer 2 Report (New Reviewer)

Comments and Suggestions for Authors

On the whole this is an interesting paper. It is relatively well structured, and given the scope of the claims made, fairly well argued.  A general concern is that I doubt it would go any way to convincing someone who was not already inclined to be sympathetic to the notion of an ecological self and the notion of environmental virtue ethics, and already fully convinced of the badness of  anthropocentrism in environmental ethics. That's probably not a huge issue given that most people reading this special issue will fall neatly into that camp. 

Some more specific comments below:

Line 10: “such a self implies that the nature…” what does the term “the nature” mean in this context? The nature of what? Or do you mean the natural environment. Please clarify.

Line 15: I would reframe the point about the distinction seeming trivial into something more positive. 

Line 16: “Finally, it allows…” what does ‘it’ refer to in the context? Does it refer to the distinction between ecological and environmental virtues? If so say “Finally, this distinction allows…”

I would  to revise the abstract so that it is shorter and more to point and also makes room for you to mention the ‘three different forms of virtue ethics’ that you intend to discuss.

Introduction

I’m not sure you can say the “environmental ethics” itself has been calling for anything. Might be better to say “many proponents of environmental ethics have been calling for..”

Why does the sentence lines 36 to 39 have a question mark at the end?

Lines 42 to 46: I’m not sure that the characterisation of virtue ethics as evaluating action as good or evil is accurate. There are virtue ethicists (ie Rosalind Hursthouse) who have attempted to give a virtue ethics based account of action guidance but most are unapologetically focused on the assessment of character, rather than of action. 

Incorrect spelling of Aristotle in footnote 5

Lines 62 to 69: The distinction between an extrinsic or intrinsic understanding of the natural environment is not clearly expressed. I would rewrite these two points to make it much more crisp.

Lines 74 to 76: The distinction between “environmental” and “ecological” has been mentioned a couple of times now, but I’m still not clear exactly what it consists in. Now would be a good time to clarify.

Lines 76 to 77: you talk about an “anthropological bias” as if being focused on human well-being is an obviously bad thing.  I know that this is a shibboleth in environmental ethics but it might be worth saying exactly what you mean by ‘anthropocentrism’ and why it should be avoided, rather than just assuming that everyone agrees.

The discussion of virtue ethics in "The Characteristic of Virtue Ethics" is mostly fine but it might be worth considering cutting it down a little bit. Likewise the discussion of Sandler's moral anthropocentrism and moral non-anthropocentrism. Also good but could be a little more succinct.

I did not find the discussion for the dictation between epistemic and ontological anthropocentrism very clear. In particular, it was not obvious how the first person perspective or the perspective of lived experience led from an 'environmental self' to the necessity to posit 'ecological self', or how this ecological self was supposed to solve the problems of an anthropocentric bias in cases where human needs might take priority over the needs/well-being/good of the non-human environment. 

Likewise some of the claims in Section 4 seemed more asserted than argued for. For instance, that an ecological self is a human — non-human relational field and not an entity... what does that even mean??

The discussion of Freya Matthews provides a slightly more concrete understanding of what the author might be getting at with the claim that a non-anthropocentric environmental virtue ethics must be centred on the flourishing of the ecological self, but I was left wondering how practical such an approach to environmental ethics might be. What would it say about the initial example posed in the paper should I take a long-haul flight to visit my ailing grandmother or refuse to because of the carbon emissions? This example discussed right at the end (regarding the management of wolves) asserts that an ecological self would "a priori" consider the legitimacy of wolves living in farming territory, but on what grounds?? I would like to see a bit more of the reasoning as to why this assertion is made, otherwise it might look like the ecological self would choose whatever happens to be the fashionable green/environmental position. What would the ecological virtue ethics position be on the conservation of the Bot Fly, for instance? and why? Would it consider "a priori" the legitimacy of the bot fly continuing to inhabit territory where humans were living. If not, why the wolf and not the bot fly?

It's an interesting paper, making a number of plausible claims but it could do with some tightening up towards the end.

Comments on the Quality of English Language

Some suggestions made above. The quality of the English Language is fine on the whole.

Author Response

Dear Reviewer,
Thank you for your critical reading of my article and your valuable comments and suggestions. You will find in the attached document the list of changes made to the article in response to your comments. The changes are highlighted in green in the manuscript.

Kind Regards

Gérald Hess

This manuscript is a resubmission of an earlier submission. The following is a list of the peer review reports and author responses from that submission.

Round 1

Reviewer 1 Report

Comments and Suggestions for Authors

Some statements about virtue ethics are inaccurate:

21-23: This is not Cafaro's view. He doesn't talk about duties or responsibilities. His aim is to identify certain character traits as "environmental virtues or excellences" - character traits that we should seek to acquire and that will allow us to flourish as human beings.

32-33: It is simply not true to say that virtue ethics places more emphasis on motivation than on action. What we can say is the following: VE evaluates actions in terms of aretaic language (e.g. as courageous, kind, etc.) rather than with reference to deontic rules (e.g. "Do not kill an innocent person). 

63-64: Again, as it stands, this statement is not true. Rather: The startingpoint for VE is the question, "What kind of human being should I become?" - which character traits should I acquire? And when thinking about character traits (virtues) we necessarily think about how we should act. So VE does seek to answer the question, "What should I do?", but it does so by thinking about the character traits that are needed for flourishing instead of abstract moral rules and principles.

65-75: There is quite a bit of some muddled writing here. Virtues are reliable dispositions to act, feel, and reason in certain ways. For a trait to be a virtue (or what makes a trait a virtue) it must contribute to human flourishing. But when acting, a virtuous person does not aim at their own happiness or flourishing. E.g., a benevolent person will aim at the good of others, an honest person will aim at discovering or revealing the truth, etc. 

It is important to see that for VE, the claim that virtues are necessary for or contribute to flourishing (a claim about what makes a trait a virtue) does not imply that the virtuous person is motivated by (or aims to produce) their own happiness, or that individual virtues are targeted at the agent's happiness.

76-82: Some confusion here, too: It is true that for Aristotle, functioning well as a human being involves rational thought and action. Thus the virtues all require reasoning well, acting for the right reasons, etc. But he does not equate the human good or flourishing with rationality.  See Kraut's entry on Aristotle's ethics in the Stanford Encyclopedia of Philosophy.

118-119: Given that virtue ethics, including Aristotle's position, is not presented accurately in this paper, it is unfair to claim that McMullin's view "renews and extends the classical - Aristotelian - tradition of virtue ethics". 

199-200: How can temperance put restraints on human happiness? In the Aristotelian view, temperance (roughly, the ability to control one's desire for pleasure derived from eating, drinking, etc.) is a virtue because it leads to human flourishing.  If you want to redefine temperance as an environmental virtue it would be by pointing out that it is necessary, not only for human flourishing but also for the good of the environment. (For useful discussion on the vice of greed, see Cafaro's chapter in Environmental Virtue Ethics).

235ff I found the rest of the paper very unclear, but this is probably because I ma not familiar with the literature in this area. I will therefore not comment on this.

Comments on the Quality of English Language

The writing is good, with only a few minor errors. 

Author Response

Dear Reviewer,

Please find in the attached document my responses to your comments on my paper.

Kind regards

Reviewer 2 Report

Comments and Suggestions for Authors

I appreciate this attempt to develop an ecological virtue ethics from a phenomenological grounding. However, you need to say more about how this approach is distinct from and superior to more common analytic approaches, such as Sandler's. See my comments on the mss.

Beyond the theoretical debate, I would like a better sense of the practical upshot. If you are right in your approach, how would that change how we think about specific ecological virtues? How would it change how we should act in the world?

Author Response

Dear Reviewer,

Please find in the attached document my responses to your comment on my paper.

Kind regards

Round 2

Reviewer 1 Report

Comments and Suggestions for Authors

The author has made some improvements, however the sections on Aristotelian virtue ethics (i.e., the sections that I am qualified to assess) remain very unclear. As noted in my previous review, the author does not demonstrate a familiarity with the literature on Aristotelian VE. For example, it is true that this position is sometimes characterised as being concerned with the question "How should I live?" and not with "What should I do?", but this characterisation has been rejected by virtue ethicists as early as Hursthouse 1999. 

A problem that is more central to the aims of this paper is that the Aristotelian view of the link between virtue and flourishing is unclear. The author doesn't seem to appreciate the difference between the question, "What makes a trait a virtue?" and "What are the aims or target of a specific virtue?" In the case of benevolence, for example, it is a virtue because it contributes to human flourishing, but the aim of benevolence is to benefit (human and non-human) others. 

Comments on the Quality of English Language

needs to be edited for clarity

Reviewer 2 Report

Comments and Suggestions for Authors

I still do not see compelling answers to my two main questions about this article: why is this approach theoretically and practically superior to other approaches to EVE, such as Sandler's?